# Average egg price: Mean reversion and persistence in a time series approach

Aitor Sarasola-Cullen[1☉], Luis Alberiko Gil-Alana [1,2☉]*

**1** Department of Economics, University of Navarra, Pamplona, Spain, **2** Faculty of Business Administration, Universidad Francisco de Vitoria, Madrid, Spain

☉ These authors contributed equally to this work.
* alana@unav.es

## Abstract

The evolution over time of the Consumer Price Index (CPI) is regarded as a key indicator of the general health and direction of any given economy. As the CPI continues to rise, the purchasing power of consumers decreases and their spending habits change significantly, making it imperative for policymakers to understand the underlying reasons that lead to such changes. A key component of the CPI basket is represented by the food and beverages items, within which eggs have undergone a significant price increase during the past years. Egg prices have a significant impact on consumers, given that eggs are a staple product, serving as the lowest cost protein alternative. This paper analyzes the long term behavior of the average egg price (cost per Dozen) in the U.S. by looking at the statistical properties of the series and using a methodology based on the concept of fractional integration. The primary goal is to determine whether the average egg price exhibits traits of long memory or mean reversion. Long memory describes the scenario where observations from a distant past have an influence on the present value of the series. Conversely, mean reversion refers to the phenomenon where data points eventually return to the long-term average after deviating from the mean for a certain period. The analysis also explores the relationship between egg prices and the Producer Price Index (PPI) through cointegration methods. Preliminary findings indicate that long memory takes place in both series and mean reversion in the PPI. Also, the two series seem to be cointegrated. This suggests the presence of a stable long-run equilibrium relationship between the Average Egg price and the Producer Price Index in the U.S., indicating a sustained co-movement between the two variables over time.

## Introduction

The evolution over time of the Consumer Price Index (CPI) is regarded as a key indicator of the general health and direction of any given economy. In the U.S. the overall CPI has increased at a steady pace over the past vicennium, experiencing a growth

**Data availability statement:** The data are available in a public repository https://github.com/Luis-cell568/Egg-prices Also in a Supporting information file.

**Funding:** Prof. Luis A. Gil-Alana gratefully acknowledges financial support from the project from 'Ministerio de Ciencia, Innovación y Universidades`Agencia Estatal de Investigación' (AEI) Spain and `Fondo Europeo de Desarrollo Regional' (FEDER), Grant PID2023-149516NB-I00 funded by MCIN/AEI/ 10.13039/501100011033. He also acknowledges support from an internal Project of the Universidad Francisco de Vitoria.

**Competing interests:** the authors have declared that no competing interests exist.

of approximately 82 percent since the year 2000. As the CPI continues to rise, the purchasing power of consumers decreases and their spending habits change significantly, making it imperative for policymakers to understand the underlying reasons that lead to such changes.

A key component of the CPI basket is represented by the food and beverages items which account for 8 percent of the index in the US and 17 percent on average in OECD countries. Given their substantial weight in the CPI, understanding the impact of price changes within the food basket is essential to interpret their effect on the CPI. In addition, due to the high volatility of food item prices, their contribution to the variation in inflation can easily exceed their weight. Thus, because of their salient features, food prices could have a disproportionate impact on inflation expectations and spread more readily to other categories [1].

Among the various items within the food and beverages category, eggs have undergone a significant price increase during the past years. According to the U.S. Bureau of Labor Statistics (BLS), eggs experienced the highest inflation rate over the 2022 period (32.2 percent) within the food and beverages group (see Fig 1). The average retail price of eggs (grade A, not seasonally adjusted) reached a record high of $4.25/dozen in December 2022, up 138% from December 2021 ($1.79/dozen) [2].

The substantial increase in egg prices observed recently has had a significant impact on consumers, given that eggs are a staple product with few substitutes, serving as the lowest cost protein alternative. Eggs are a key component in the standard American's diet with the US table-egg production estimated at around 635.0 million dozen eggs in November 2022 and an average layer flock size of 308.8 million layers. In addition, it is estimated that exports of eggs and egg products in November 2022 reached 18.0 million dozen (shell-egg) equivalent [3].

This paper analyzes the long term behavior of the average egg price (cost per Dozen) in the U.S. by looking at the statistical properties of the series and using a methodology based on fractional integration. The primary goal is to determine whether the average egg price exhibits traits of long memory or mean reversion. Long memory describes the scenario where observations from a distant past have an influence on the present value of the series. Conversely, mean reversion refers to the phenomenon where data points eventually return to the long-term average after deviating from the mean for a certain period.

By using fractional integration techniques, we are able to discern whether the effects of shocks on the data will be transitory or permanent. This methodology is highly appropriate for persistence analysis and evaluating the nature of shocks, being more flexible and general than the traditional methods that focus exclusively on integer degrees of differentiation [4]. In essence, fractional integration enables the differencing parameter, say "$d$", to take any real value, including fractional ones. By estimating the differencing parameter $d$, we are able to discern whether the series exhibits long memory [$d > 0$] or short memory [$d = 0$], as well as whether the series will exhibit reversion to the mean [$d < 1$]. Further details regarding the methodology will be provided throughout the paper.

To conduct our analysis, we will examine data encompassing the average egg price throughout the time period spanning from 1980 to 2023 (see Fig 2).

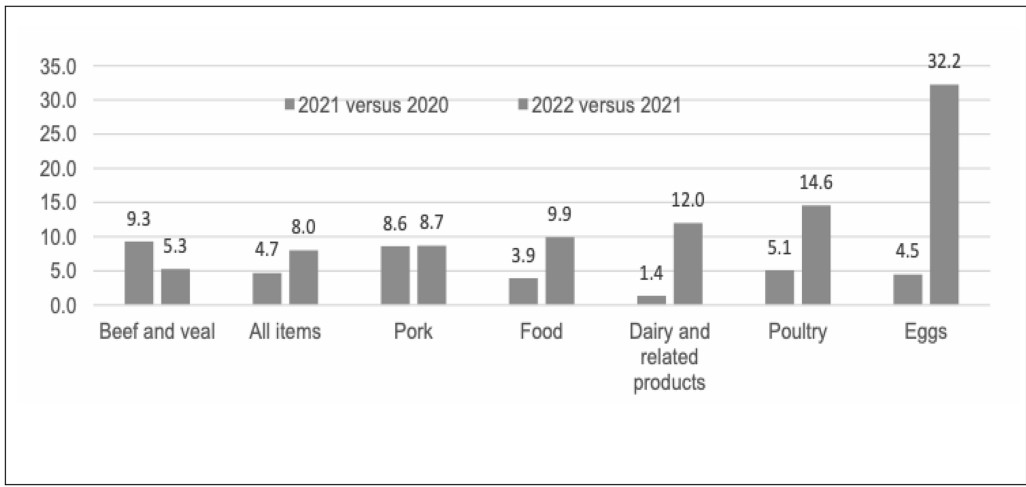

**Fig 1. Livestock, Dairy, and Poultry Outlook, Percent increase over last year.** Source: United States Department of Agriculture (2023), "Livestock, Dairy, and Poultry Outlook: January 2023". Economic Research Service retrieved from FRED, Fderal Reserve Bank of St. Louis. Available at: https://www.ers.usda.gov/webdocs/outlooks/105645/ldp-m-343.pdf?v=4870.7.

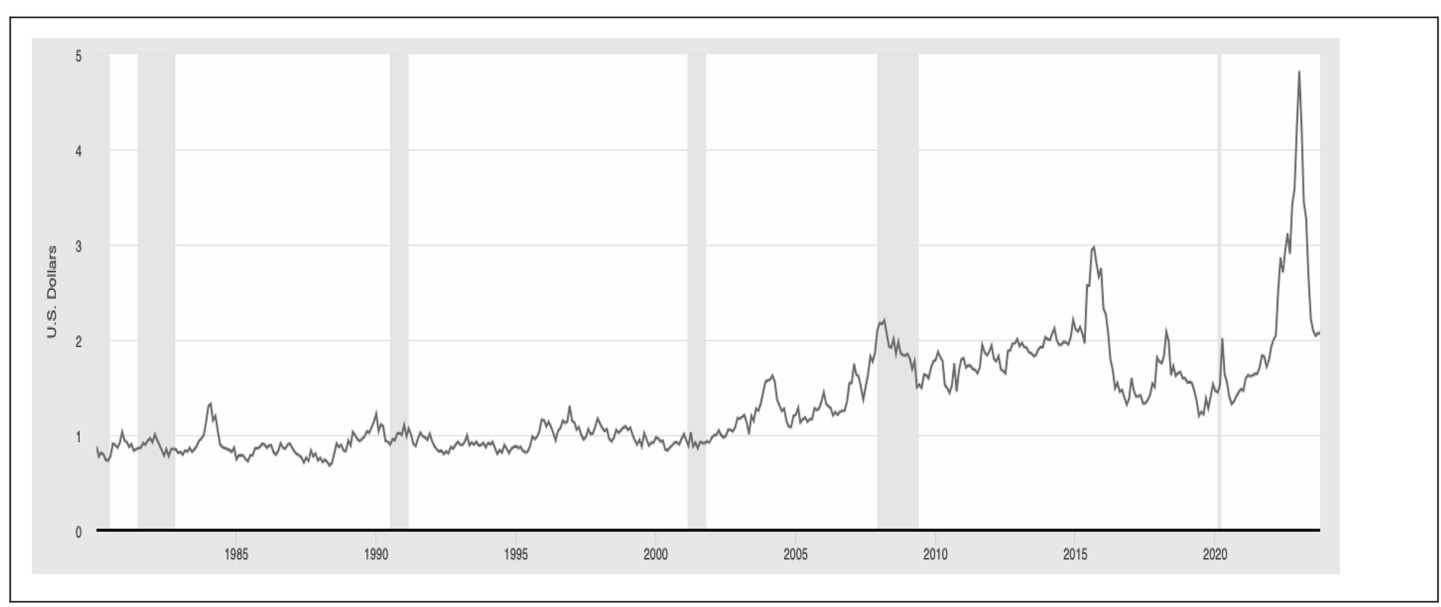

**Fig 2. Average egg price, Grade A, large (Cost per dozen) in U.S. city average. Source:** U.S. Bureau of Labor Statistics, Average Price: Eggs, Grade A, Larg (Cost per Doze) in U.S. City Average [APU0000708111], retrieved from FRED, Federal Reserve Bank of St. Louis. Available at: https://fred.stlouisfed.org/series/APU000708111.

Furthermore, we plan to understand whether there is a relationship between the average egg price and the Producer Price Index by applying cointegration techniques. The Producer Price Index (PPI) measures the average change over time in the selling prices received by domestic producers for their output [2]. The prices considered in the PPI are obtained from the initial commercial transaction, with the exclusion of any taxes, transportation expenses, and trade margins that the buyer may be required to bear.

Examining the PPI is interesting due to its diverse applications. First, it serves as an indicator of overall price shifts at the producer level, capturing price movements before reaching the retail stage. Additionally, it holds significant utility for the U.S. Presidency, Congress, and the Federal Reserve when shaping fiscal and monetary policies, serving as a valuable tool for comparing input and output costs. Finally, another application of PPI involves contract adjustments, where PPI data is commonly employed to adjust purchase and sales contracts in response to fluctuations in input prices.

An examination of the time series properties of the Average Egg Price is particularly relevant given the recent price increase. According to certain assessments, the increase in price is projected to continue in the long term driven by supply-chain disruptions due to the COVID-19 pandemic, increased egg demand and overall inflation. However, others argue that the volatility of the series will return to the long term average over time (mean reversion), making it ineffective for policymakers to take proactive measures.

To fully understand the data at hand, it is crucial to examine the qualitative reasons that have contributed to the recent spike in prices. Analyzing such reasons will enable us to better understand the data properties and assess whether such findings are coherent with our results. The recent surge in egg prices can be attributed to a confluence of factors including supply-chain disruptions due to the COVID-19 pandemic, overall inflation, increased egg demand during the holiday season and the recent outbreaks of highly pathogenic avian influenza (HPAI) (i.e., bird flu). In addition, the increase in worldwide animal feed prices as a result of the ongoing war in Ukraine has had a significant effect on egg prices.

Regarding the supply chain disruptions due to the COVID-19 pandemic, it is worth highlighting that the egg demand experienced a significant uptick during the pandemic due to the shift from food-away-from-home to food-at-home consumption. While grocery demand experienced a surge, the egg supply chain initially struggled to adapt and accommodate large volumes for the retail market. Consequently, there was a significant price increase due to low supply, leading to approximately 141% and 182% increases in retail and farm-gate prices for table eggs, respectively [5]. Since then, the egg industry's supply chain has faced challenges in meeting the increased demand, particularly with the recent reduction in flock size due to the highly pathogenic avian influenza (HPAI). Other factors such as the generalized increase of inputs in egg production, including natural gas and feed costs, along with the seasonal increase in demand during the winter and spring, driven by the Easter and Christmas holiday periods, have adversely affected egg prices.

Avian influenza is a disease that primarily affects birds and is caused by a virus of the Orthomyxoviridae family. According to the Pan American Health Organization, the Low Pathogenic Avian Influenza Virus (LPAIV) can cause a mild illness, often unnoticed or without any symptoms. Conversely, the Highly Pathogenic Avian Influenza Virus (HPAIV), the variant impacting the U.S. commercial table egg layer flock, causes serious illness in birds that can spread rapidly, resulting in high death rates in different species of birds [6]. Due to its highly contagious nature and the potential risk to humans, the USDA takes measures to control the spread of HPAI by euthanizing both infected birds and affected flocks. Thus, avian influenza outbreaks have significant impacts on commercial bird populations in affected areas [7] and the egg production in the US.

With regards to the recent outbreak, the first reported HPAI in the U.S. commercial table egg layer flock was on February 22, 2022, in New Castle County, Delaware, affecting over a million birds. Since then, the disease and related depopulation protocols have affected over 69.5 million commercial layers, impacting over 47 states across the U.S. (Fig 3).

Besides having detrimental effects through bird losses, outbreaks have a significant impact on the overall flock size of egg production and, ultimately, the price paid by consumers. For instance, due to HPAI outbreaks in 2022, the table egg laying flock shrank to 299 million birds, while the average egg price rose from $2.005 in February 2022 to $4.211 in February 2023. Since the first HPAI outbreak was reported in February 2022, the monthly size of the egg-type layer flock has been, on average, 4.8 percent year-over-year lower, while table-egg production was, on average, 3.2 percent year-over-year lower [3].

Having understood the qualitative reasons that have led to the recent surge in egg prices, we analyze the long term behavior of the average egg price (cost per Dozen) in the U.S. by looking at the statistical properties of the series and

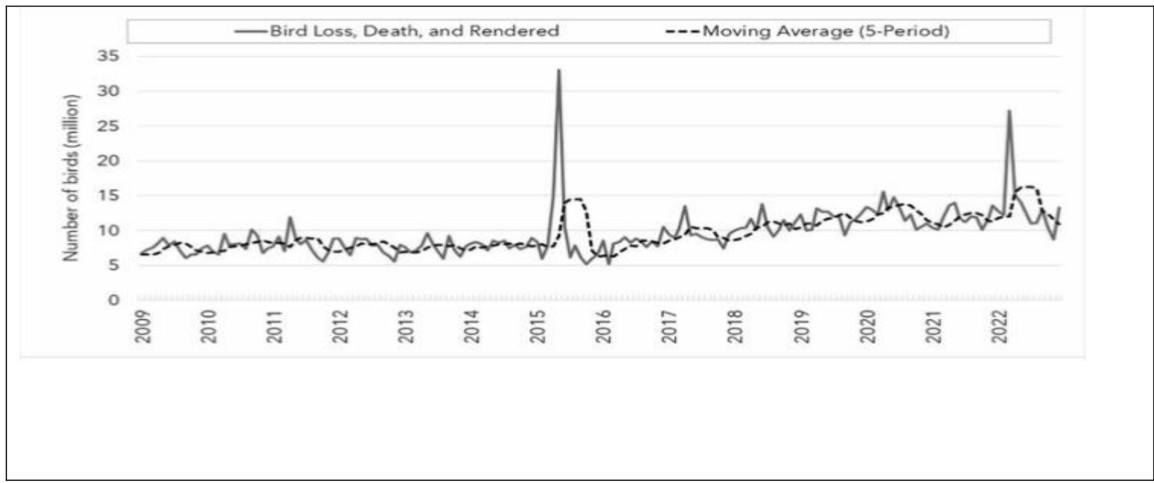

**Fig 3. Monthly bird losses for all commercial layers in the U.S. Source:** U.S. Department of Agriculture, 2023c. Quick Stats. Washingtong, DC: USDA National Agricultural Statistics Service. Available at: https://quickstats.nass.usda.gov/.

using fractional differentiation. In this context, we focus our attention on items such as the existence of trends, mean reversion, non-stationarity and other features of the data.

## Literature review

There are numerous studies that have focused the attention in the modelization of food prices throughout different techniques. Thus, for example, Mao et al. [8] investigated China's vegetable prices by using ARIMA models in order to determine the factors that influence vegetable price fluctuations. Jayatri [9] used a more extended ARIMA-GARCH model to examine the volatility in various essential food prices in Indonesia. In a similar vein, Magnogna et al. [10] employed a hybrid long short-term memory model for the analysis of agricultural price volatility in India. Other studies have performed empirical studies on price forecasting of individual agricultural prices such as soybean [11], pork [12,13], corn [14] and beef [15]. Nevertheless, in the economic literature, the statistical properties of the average egg price have received limited attention, with existing studies frequently relying on standard unit root approaches based on deterministic (usually, functions of time) regressions. In earlier studies, the fundamental issue was to determine whether the evolution of egg prices is better characterized as a stationary process I(0) or a nonstationary I(1) one. In this context, Rumánková [16] found out that most agricultural time series were non-stationary but stationary after first differentiation. Zhang et al. [17] used a STVAR model to identify asymmetric impacts of market conditions on multi-scale systemic risk spillovers of commodity market. On the other hand, Pechrová and Šimpach [18] conducted a study to model the development of consumer egg prices for the Czech Republic. In their study, the differentiation of the first order caused the process to be only a white noise. Despite facing the challenge of modelling this process, they ultimately opted for first-order differentiation, setting aside fractional integration methods.

Another study regarding forecasting egg prices was done by Raghavender and Udayshankar [19] in Telangana, India. In this study the objective was to analyze the seasonal price variations of eggs and to develop a seasonal autoregressive integrated moving average (SARIMA) model to forecast the average price of eggs in Hyderabad in Telangana state. Throughout the investigation, fractional integration techniques were not utilized, and standard practices were applied instead. Although it has been proved that differencing time series data consistent with *I(1)* processes could lead in some cases to overdifferencing at the zero frequency [20], no studies using fractional integration techniques have been found

regarding any egg market. Thus, in this work, we take a general approach from a fractionally integrated perspective. This methodology enables a much richer degree of flexibility in the dynamic behavior of the series than the traditional approaches [21]. Moreover, the fractionally integrated or I(d) models can help us discern whether shocks in the series have transitory, either with exponentially fast decays (i.e., I(0)) or hyperbolically slow decays (I(d), $0 < d < 1$) versus permanent or even explosive patterns (I(d), $d \geq 1$) [22]. Thus, in the former case mean reversion takes place and shocks will tend to disappear by themselves in the long run implying lack of policy actions to recover the original level of prices. On the other hand, on the presence of unit roots or explosive patterns shocks will have a permanent nature and strong policy actions will be required to recover the original prices. Note that the main advantage of the fractional integration framework is that it allows any real value for the degree of differentiation including then the particular cases of stationarity I(0) and nonstationarity I(1) as particular cases of interest of our approach. Recent examples of empirical applications of fractional integration on inflation rates include among others the paper by Usman and Gil-Alana [23] and Solarin et al. [24] and other applications dealing with persistence in commodity prices are Alfeus and Nikitopoulos [25], Gil-Alana and Font de Villanueva [26] and Basistha and Startz [27].

## Descriptive analysis

The time series data analyzed in this study corresponds to the average egg price in the U.S. market. The data includes the monthly cost per dozen of Grade A eggs, covering the period from 1980 to 2023. This information is obtained from the Federal Reserve Bank of St. Louis, and utilizes data directly from the U.S. Bureau of Labor Statistics (BLS), collected monthly by BLS representatives in 75 urban areas for the Consumer Price Index (CPI).

Table 1 summarizes some descriptive statistics for both the average egg price and the PPI in the U.S. We observe that the highest average egg price was recorded in 2023-01-01, 11 months after the first reported HPAI in the U.S. The HPAI devastated poultry flocks across the country, pushing egg prices to record highs, an issue exacerbated by inflation and persistent supply chain disruptions. During this period, egg production measured by the dozen decreased by 6.45%, dropping from 806,416,200 in January 2022–754,374,800 in January 2023. Conversely, the lowest average egg price was recorded in 1988-05-01, with a value of $0.678. Furthermore, the average egg price during the 43 year period is set at 1.328, approximately a quarter of the price recorded in January 2023. In addition, the standard deviation, which indicates the volatility of the series, is set at around 0.424. This suggests that the average egg prices exhibit relatively limited fluctuations around the mean, indicating a certain degree of stability in the pricing pattern.

(PPI), much like the average egg price, data is acquired through the U.S. Bureau of Labor Statistics (BLS). The BLS utilizes a wide sample encompassing over 16,000 establishments, resulting in approximately 64,000 price quotations per month. Establishments are selected for the PPI survey via systematic sampling of a list of all firms in the industry. These chosen establishments voluntarily contribute price data via a secure online platform.

In terms of descriptive statistics, it is worth noting that the highest Producer Price Index (PPI) reached a value of $580.66, documented on December 1, 2022. This peak coincides with the Christmas period and was observed 10 months after the initial outbreak. This highest PPI value occurred one month prior to the peak in the Average Egg Price, suggesting a potential correlation between these two variables. On the other hand, the lowest PPI was recorded

**Table 1. Descriptive statistics.**

| Index | Max | Min | Average | St.Dev. | St.Dev/Average |
|---|---|---|---|---|---|
| Average Egg Price | 4.823 | 0.678 | 1.328 | 0.564 | 0.424 |
| PPI Egg Price | 580.662 | 61.500 | 125.120 | 63.160 | 0.500 |

Source: own sources.

in 1999-10-01, with a value of \$61.50. Other interesting statistics include the average PPI (\$125.12) and the standard deviation (63.160).

To assess variation across various mean scales we decided to use the coefficient of variation (CV), calculated as the standard deviation divided by the mean. It measures the variability of a series of numbers independently of the unit of measurement used for these numbers [28]. A higher coefficient of variation indicates a greater level of dispersion around the mean. As we can observe, the coefficient of variation in both series is relatively similar, indicating that both series have a similar dispersion around the mean.

Data and codes used in this application are available in the following repository 10.5281/zenodo.17415399 at https://zenodo.org/uploads/17415399.

## Methods

The primary goal is to determine whether the average egg price series exhibits long memory and/or mean reversion. To do so, we first need to define short memory, also known as integrated of order 0 or I(0). Mathematically, given a covariance stationary process $\{x(t), t = 0, \pm 1, \ldots\}$ with mean $E(x(t)) = \mu$, it is stated that it displays the property of short memory if the infinite sum of the autocovariances, denoted as $\gamma(u) = E[(x(t) - \mu)(x(t+u) - \mu)]$, is finite, that is,

$$\sum_{j=-\infty}^{\infty} |\gamma(j)| < \infty. \tag{1}$$

Alternatively, short memory can be defined in the frequency domain by introducing the spectral density function, $f(\lambda)$, that is the Fourier transformation of the autocovariances, i.e.,

$$f(\lambda) = \frac{1}{2\pi} \sum_{j=-\infty}^{\infty} \gamma(j) e^{i\lambda j} = \frac{1}{2\pi} \left( \gamma(0) + 2 \sum_{j=1}^{\infty} \gamma(j) \cos(\lambda j) \right). \tag{2}$$

In this context, $x(t)$ is short memory if the spectral density function is positive and bounded at all frequencies, i.e.,

$$0 < f(\lambda) < \infty. \qquad \text{for all } \lambda \in [0, \pi). \tag{3}$$

Within this category, one encounters not only the white noise model but also the stationary and invertible AutoRegressive Moving Average (ARMA) class of models whenever there is a sort of time (weak) dependence structure.

On the other hand, a process is said to be long memory if the infinite sum of autocorrelations becomes unbounded, i.e.,

$$\sum_{j=-\infty}^{\infty} |\gamma(j)| = \infty, \tag{4}$$

or, alternative, using the frequency domain definition, if the spectral density function tends to infinity at least at one point in the frequency $[0, \pi)$,

$$f(\lambda) \to \infty, \text{ for some } \lambda \in [0, \pi). \tag{5}$$

One model satisfying the two properties in (4) and (5) is the one founded on the idea of fractional integration or integration of order d, also denoted as I(d). A process $x(t)$ is said to be I(d) if it can be expressed as:

$$(1 - L)^d x(t) = u(t), \qquad t = 0, \pm 1, \ldots. \tag{6}$$

where $L$ is the lag operator, i.e., $Lx(t) = x(t-1)$ and $u(t)$ is $I(0)$. Then, as long as $d$ is positive and smaller than 0.5, $x(t)$ in (6) becomes long memory as the infinite sum of the autocovariances becomes infinite. Alternatively, this feature can be noticed as its spectral density function becomes:

$$f(\lambda) = \frac{\sigma^2}{2\pi}\left|\frac{1}{1-e^{i\lambda}}\right|^d ,$$

(7)

and it tends to infinity as $\lambda \to 0^+$ when $d>0$. Using a binomial expansion, the polynomial in $L$ in (6) can be expressed as:

$$(1-L)^d = \sum_{j=0}^{\infty} \binom{d}{j}(-1)^j L^j = 1 - dL + \frac{d(d-1)}{2}L^2 - \frac{d(d-1)(d-2)}{6}L^3 \cdots ,$$

(8)

implying that

$$x(t) = dx(t-1) - \frac{d(d-1)}{2}x(t-2) + \frac{d(d-1)(d-2)}{6}x(t-3) - \cdots + u(t),$$

In this context, the differencing parameter $d$ can be interpreted as a measure of the degree of persistence of the data. As the value of "$d$" increases, the level of persistence grows, resulting

in a stronger association between observations, even if they are far apart in time. By allowing $d$ to be a fractional value, we can consider a wide range of alternatives, including;

i)   anti-persistence (if $d<0$),

ii)  short memory (if $d=0$),

iii) stationary long memory with mean reversion (if $0<d<0.5$),

iv) nonstationary with mean reversion (if $0.5\leq d<1$),

v)  unit roots (if $d=1$),

and

vi) explosive patterns (if $d>1$).

From a policy making perspective, the crucial value to look for is 1, since mean reversion and transitory nature only occurs whenever $d$ is strictly smaller than to such value. On the contrary, if $d$ is equal to or higher than 1, there is lack of mean reversion and thus permanency of shocks, making it imperative for policy makers to take action.

Not only does fractional integration allow us to analyze the level of persistence on the series, but it also addresses the issue of over-differentiation in aggregated nonstationary $I(1)$ processes by permitting a fractional degree of differentiation below 1. By doing so the conventional practice of taking first differences was much improved, as the fractional methodology proved to be more flexible and general than the traditional methods.

The estimation relies on the parametric approach of Robinson [29], which is a testing procedure based on the Lagrange Multiplier (LM) principle and that uses the likelihood function in the frequency domain. We use here a very simple approach of this method that has been widely used in the empirical literature (see [30] for the specific functional form of the tests used in this application, whose codes can be found in [31]). The main features of this procedure are that it is valid for any real $d$ and thus it does not impose preliminary differentiation in case of nonstationary data; moreover, it has a standard normal limit distribution and it is the most efficient method in the context of local alternatives. Nevertheless, as

a robustness check, we also employed alternative approaches like the parametric method of Sowell [32] and the semi-parametric log-periodogram estimate of Geweke and Porter-Hudak [33] and others like Robinson [34] and Velasco [35], all producing very similar results to those reported in this work.

## Results and discussion

Following standard parameterization, we incorporate a linear time trend of the form:

$$y(t) = \alpha + \beta t + x(t), \qquad t = 1, 2, \ldots . \tag{9}$$

where α and β represent coefficients that need estimation—namely, a constant and a time trend, respectively. The variable x(t) is determined by equation (6). In other words, the model under examination can be expressed as,

$$y(t) = \alpha + \beta t + x(t), \quad (1 - L)^d x(t) = u(t), \qquad t = 1, 2, \ldots . \tag{10}$$

where u(t) is I(0) or a short-memory process. We then proceed to make different assumptions in relation to the u(t) term. Thus, we first suppose that u(t) is a white noise process. Then, we allow some type of time dependence; however, instead of using the classical ARMA approach we use the exponential spectral model of Bloomfield [36] which approximates ARMA structures with very few parameters. Finally, we use a seasonal AR(1) model to take into account the potential seasonal structure of the data.

The results reported across Tables 2 and 3 encompass the estimates of the differencing parameter d along with the 95% confidence bands for the estimates under three different model specifications, namely i) with no deterministic terms, i.e., imposing $\alpha = \beta = 0$ in (10) (results displayed in column 2); ii) with a constant, i.e., $\beta = 0$ in (10) (results displayed in column 3); and iii) with a constant and a linear time trend (results displayed in the last column).

The coefficients marked in bold in the table are those from the model selected in each case on the basis of the statistical significance of the coefficients (unreported). In other words, when both coefficients α and β are statistically significant, we opt for that model with a constant and a linear time trend, and the estimates of d are highlighted in bold in column 4 of the tables. Conversely, if the β-coefficient is not significant, we shift to the model with an intercept, showcasing the values of d in bold within column 3 of the tables. Ultimately, if both coefficients are statistically insignificant, we select the model with no deterministic terms. The upper section of the tables corresponds to the original data, whereas the lower section pertains to the logged values.

**Table 2. Estimates of d in a model given by (10). Series: Average Egg Price.**

**i) Original data**

| Disturbances | No terms | An intercept | An intercept with a linear time trend |
|---|---|---|---|
| White noise | 1.06 (0.98, 1.15) | **1.09 (1.01, 1.18)** | 1.09 (1.01, 1.18) |
| Bloomfield | 0.91 (0.78, 1.11) | **0.99 (0.81, 1.20)** | 0.99 (0.80, 1.20) |
| Seasonal AR(1) | 1.06 (0.98, 1.15) | **1.10 (1.02, 1.19)** | 1.10 (1.02, 1.19) |

**ii) Logged data**

| Disturbances | No terms | An intercept | An intercept with a linear time trend |
|---|---|---|---|
| White noise | 0.97 (0.90, 1.04) | **0.96 (0.89, 1.04)** | 0.96 (0.89, 1.04) |
| Bloomfield | 0.99 (0.83, 1.17) | **0.96 (0.83, 1.17)** | 0.96 (0.82, 1.17) |
| Seasonal AR(1) | 0.98 (0.92, 1.06) | **0.97 (0.90, 1.05)** | 0.97 (0.90, 1.05) |

**Note:** The sample period goes from January 1980 until October 2023 (526 observations). Column 2 reports the results in a model with no deterministic terms. Those in column 3 refers to the model incorporating an intercept, while those in column 4 are those corresponding to a model with an intercept and a linear time trend. The values in parenthesis contain the 95% confidence bands for the differencing parameter. In bold the selected deterministic model for each series.

**Table 3. Estimates of d in a model given by (10). Series: Producer Price Index.**

**i) Original data**

| Disturbances | No terms | An intercept | An intercept with a linear time trend |
|---|---|---|---|
| White noise | 0.69 (0.62, 0.78) | **0.71 (0.63, 0.79)** | 0.70 (0.62, 0.79) |
| Bloomfield | 0.68 (0.51, 0.88) | **0.69 (0.54, 0.90)** | 0.67 (0.51, 0.90) |
| Seasonal AR(1) | 0.69 (0.62, 0.78) | **0.70 (0.63, 0.79)** | 0.70 (0.62, 0.79) |

**ii) Logged data**

| Disturbances | No terms | An intercept | An intercept with a linear time trend |
|---|---|---|---|
| White noise | 0.85 (0.80, 0.93) | **0.65 (0.58, 0.73)** | 0.64 (0.57, 0.72) |
| Bloomfield | 0.93 (0.82, 1.06) | **0.71 (0.59, 0.91)** | 0.73 (0.54, 0.91) |
| Seasonal AR(1) | 0.85 (0.79, 0.90) | **0.65 (0.58, 0.73)** | 0.64 (0.57, 0.72) |

**Note:** The sample period goes from January 1980 until October 2023 (526 observations). Column 2 reports the results in a model with no deterministic terms. Those in column 3 refers to the model incorporating an intercept, while those in column 4 are those corresponding to a model with an intercept and a linear time trend. The values in parenthesis contain the 95% confidence bands for the differencing parameter. In bold the selected deterministic model for each series.

With regards to the results reported, we observe that in both tables the time trend coefficient ($\beta$) is not found to be statistically significant in any single case; whereas the intercept is significant in all cases.

Moving to the Average Egg Price, in Table 2, we observe that the estimates of $d$ are equal to or higher than 1 in all cases. Thus, for the original prices, d is found to be above 1 if the residuals are a white noise or seasonal AR, indicating explosive patterns (as $d > 1$ and the unit root null can be rejected); however, if $u(t)$ is autocorrelated throughout the model of Bloomfield [36], the unit root null, i.e., $d = 1$ cannot be rejected. As a consequence, when $u(t)$ is autocorrelated, we are not able to conclude that neither mean reversion nor explosive pattern take place as the confidence interval ranges from (0.83, 1.17). For the logged data, the $I(1)$ hypothesis cannot be rejected for any type of disturbances, and the estimate of $d$ is equal to 0.96 with white noise and Bloomfield, and 0.97 with seasonal AR disturbances.

In the context of the Producer Price index data, the values are much lower and mean reversion seems to take place in all cases. Here, the $I(1)$ hypothesis can be rejected in all cases as none of the confidence bands encompasses the value of 1. The estimates of d are equal to 0.71 (white noise), 0.69 (Bloomfield) and 0.70 (seasonal AR) for the original data, and for the logged data, the values are 0.65 (white noise and seasonal AR) and 0.71 with Bloomfield disturbances. The fact that the orders of integration are in the interval [0.5, 1) indicate that the process of reversion is slow compared with the standard case of I(0) errors. Though not reported in the manuscript, we also conducted some diagnostic check on the estimated residuals on the $d$-differenced series; in particular, we implemented various Portmanteau tests [37,38] and the Lagrange Multiplier (LM) test of Breusch and Godfrey [39,40], and the results reported evidence of no additional serial correlation in the data.

Based on the estimated parameters, it is reasonable to conclude that fluctuations in average egg prices will have long lasting (or even permanent) effects in the U.S. since the unit root null hypothesis cannot be rejected under any of the proposed scenarios, while changes in the Production Price Index tend to return to the long-term average over time, though the process is relatively slow. Note that seasonality does not appear as a relevant issue in any of the two series investigated since the estimates of $d$ are practically identical under the seasonal AR specification and the white noise form for the error term. In fact, though unreported, the seasonal AR coefficients were statistically insignificant in the two cases. It should finally be noted that several Ljung-Box Q-test statistics were conducted on the residuals from the white noise and seasonal AR(1) specifications and the results suggested that they were no additional serial correlation in the data.

The next step involves checking for the possible existence of a relationship between the variables Average Egg Price and the Producer Price Index. To do so, we first compute the correlation coefficient between the two variables, which

serves as a statistical measure of the strength of a linear relationship between them. Performing the correlation coefficient, it stands at 0.89, which indicates a meaningful positive correlation.

These findings align with reality, as an increase in the Producer Price Index—representing selling prices received by domestic producers for their output—positively correlates with the average egg price paid by consumers. In the egg industry, much like in others, when output prices increase, such expenses are generally reflected in the final prices paid by consumers.

Although the existence of a relationship between the two variables may seem obvious, for a more rigorous examination of the presence of a long-run relationship we use fractional cointegration methods. One of the recent advancements in the literature on (fractional) cointegration is the Fractionally Cointegrated Vector Autoregressive (FCVAR) model introduced by Johansen [41] and further discussed by Johansen and Nielsen [42,43]. This model generalizes Johansen's [44] cointegrated vector autoregressive (CVAR) model to allow for fractionally integrated time series that cointegrate to a lower (fractional) order [45].

However, prior to analyzing the existence of a long-run relationship between the two variables, it is important to perform a causality test [46]. The test is based on the frequency domain and uses a bivariate vector autoregressive (VAR) model to construct a simple test procedure that is based on a set of linear hypotheses on the autoregressive parameters. This test procedure can easily be generalized to allow for cointegration relationships and higher-dimensional systems.

As mentioned earlier, it is recommended to examine causality before engaging in cointegration analysis. This precaution is essential because if no causality is established, we would consider the relationship statistically "spurious." In such a case, interpreting the results of the FCVAR model would be irrelevant.

Table 4 presents the results of the causality test in the frequency domain based on Breitung and Candelon [46] for the time series in both directions. Looking at the results we can conclude that the Producer Price Index (PPI) causes effects on egg prices in the long, medium, and short term. On the other hand, focusing on the statistics of the Wald test and p-values (in parentheses) for the second hypothesis, the causality test reveals a medium and short-term impact of egg prices on the Producer Price Index.

Given the established causality between the two variables, our next step involves conducting cointegration analysis. For this purpose, we use the FCVAR model, known for its numerous advantages in estimating a system of fractional time series variables that are potentially cointegrated. The flexibility of the model permits one to determine the cointegrating rank, or number of equilibrium relations, via statistical tests and to jointly estimate the adjustment coefficients and the cointegrating relations, while accounting for the short-run dynamics [45].

Table 5 displays the results obtained using the FCVAR model for both directions (eggs prices vs PPI; and PPI vs eggs prices, respectively). Our focus centers on the integrating and cointegrating parameters, which helps us analyze the long-term relationship between the two series. To grasp the findings accurately, it is essential to consider the definition of cointegration in a bivariate context as outlined by Engle and Granger [47]. They define cointegration as a situation where

**Table 4. Breitung and Candelon frequency domain causality test results.**

| Hypothesis | Long Term ($\omega = 0.05$) | Medium Term ($\omega = 1.5$) | Short Term ($\omega = 2.5$) |
|---|---|---|---|
| PPI over Average Egg Price | 248.21* (0.00) | 128.76* (0.00) | 132.33* (0.00) |
| Average Egg Price over PPI | 4.24 (0.12) | 56.66* (0.00) | 43.13* (0.00) |

**Note:** * Shows that there is a significant causality relationship at the 5% significance level. The values in the brackets are the probability value of the F statistics calculated for the relevant ω values.

**Table 5. FCVAR results.**

| Time Series | $d \neq b$ |
|---|---|
| Average Eggs Price vs PPI | $d$ = 0.723 (0.065)<br>$b$ = 0.723 (0.077) |
| | $\Delta^d \left( [Av.Eggs.Pr.\ PPI\ ] - [0.998\ 93.474\ ] \right) = L_d[0.049\ 26.757\ ]\nu_t + \sum_{i=1}^{2} \hat{\Gamma}_i \Delta^d L_d^i (X_t - \mu) + \varepsilon_t$ |
| PPI vs Average Eggs Price | $d$ = 0.723 (0.064)<br>$b$ = 0.723 (0.077) |
| | $\Delta^d \left( [PPI\ Av.Eggs.Pr.\ ] - [93.474\ 0.086\ ] \right) = L_d[-0.404\ \ -0.001\ ]\nu_t + \sum_{i=1}^{2} \hat{\Gamma}_i \Delta^d L_d^i (X_t - \mu) + \varepsilon_t$ |

**Note:** The fractional CVAR (FCVAR) approach of Johansen and Nielsen [42,43] is performed here on average egg prices and PPI. The sample period goes from January 1980 until October 2023 (526 observations).

the two individual series are integrated of order d (i.e., *I(d)*), but there exists a linear combination of the two that is integrated of a smaller order, denoted as $d - b$, where $b > 0$.

Upon examining the integrating and cointegrating parameters, it becomes apparent that for both scenarios, the order of integration for each series and the reduction in the order in the cointegrating regression stand at 0.723 ($d = b = 0.723$). These results imply *I(0)* cointegrating errors, as there is a relationship between the two series of order 0.723–0.723 = 0. Such results indicate that there is evidence of a stable long-run equilibrium relationship between the Average Egg price and the Producer Price Index in the U.S., implying long run co-movement between the two variables.

## Conclusions

The objective of this paper has been to analyze the stochastic behavior of the average egg price (cost per dozen) in the U.S. by looking at the statistical properties of the series and using fractional integration. Additionally, the study aimed to understand the possible long run relationship between PPI and the average egg price.

Analyzing such a topic was deemed relevant due to the significant size of the U.S. egg market, the influence of eggs on the food and beverages category within the Consumer Price Index (CPI), and, especially, the pivotal role of eggs in the human diet. Moreover, the study provided insights into the nature of the recent peak in the average egg price, examining whether it was necessary or not for policymakers to take action over it. Lastly, this paper aims to address the limited existing literature regarding the statistical properties of the average egg price.

The results indicate that in both the average egg price and the PPI, the selected model specification includes an intercept but not a time trend. With regards to the average egg price, the unit root null hypothesis, i.e., $d = 1$ is rejected under the white noise and seasonal AR models as $d$ is much higher than 1, indicating explosive patterns, whereas for the Bloomfield [36] model such hypothesis cannot be rejected. Nevertheless, in both cases we observe lack of reversion to the mean, implying permanency of shocks and requiring policy actions (e.g., subsidy design or inventory stabilization) in case of shocks to recover the original trends. Conversely, for the Producer Price index data, mean reversion seems to take place in all cases, making it useless for policymakers to take action when sudden peaks occur.

The primary discovery of this paper supports the notion that disturbances in average egg prices will have enduring effects in the U.S., as none of the series displays short memory traits (i.e., *I(0)*). Thus, long memory takes place in all cases with positive values of d across all confidence intervals. This suggests that abrupt changes or fluctuations in prices are likely to have a more prolonged impact. Moreover, the fact that the intervals include the value of d = 1 indicates that we cannot reject the hypothesis of a unit root in the series and therefore rejecting the hypothesis of reversion to the mean. On the other hand, regarding the Producer Price Index, despite the absence of short memory traits, the parameter *d* falls

within the range $0.5 \leq d < 1$. This indicates that the time series is nonstationary with mean reversion, implying that volatility tends to revert to a long-run average over time.

Furthermore, both the correlation coefficient and the causality test in the frequency domain based on Breitung and Candelon [46] prove the existence of some kind of relationship between the two variables. The relationship between the two variables is positive; this idea is supported by both the correlation coefficient and the qualitative analysis done throughout the analysis. Nevertheless, it should be taken into account that this relationship (Average Egg Price→PPI) is only significant in the short and medium term but not in the long term, which may be undermining symmetry and the cointegration relationship.

Finally, according to the results obtained using the FCVAR model, we can conclude that the average egg price and the producer price index are cointegrated with the cointegrating errors displaying an *I(0)* behaviour. These findings suggest the presence of a stable long-run equilibrium relationship between the Average Egg price and the Producer Price Index in the U.S., indicating a sustained co-movement between the two variables over time.

The paper contains some limitations. First, the possibility of structural breaks is an issue that has not been investigated in the present work. This is relevant since long memory and fractional integration are closely related to the issue of breaks (see, [48,49]; etc.). In this context, the methodology proposed in Gil-Alana [50], which is basically an extension of Bai and Perron's [51] approach to the fractional case can be implemented to the same data used in this application. In addition, non-linear trends can replace the linear structure employed in this work. Other alternative long memory model can also be used.

## Supporting information

**S1 Data.**
(XLS)

## Acknowledgments

Comments from the Editor and three anonymous reviewers are gratefully acknowledged.

## Author contributions

**Conceptualization:** Luis Alberiko Gil-Alana.

**Data curation:** Aitor Sarasola-Cullen.

**Formal analysis:** Aitor Sarasola-Cullen, Luis Alberiko Gil-Alana.

**Funding acquisition:** Luis Alberiko Gil-Alana.

**Methodology:** Aitor Sarasola-Cullen.

**Software:** Luis Alberiko Gil-Alana.

**Supervision:** Aitor Sarasola-Cullen, luis Alberiko Gil-Alana.

**Validation:** Aitor Sarasola-Cullen, Luis Alberiko Gil-Alana.

**Visualization:** Aitor Sarasola-Cullen.

**Writing – original draft:** Luis Alberiko Gil-Alana.

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
