## [Decision Letter · Decision Letter 0]

2 Jul 2025

Dear Dr. Gil-Alana,

Thank you for submitting your manuscript to PLOS ONE. After careful consideration, we feel that it has merit but does not fully meet PLOS ONE’s publication criteria as it currently stands. Therefore, we invite you to submit a revised version of the manuscript that addresses the points raised during the review process.

We look forward to receiving your revised manuscript.

Kind regards,

Muhammad Umer Arshad

Academic Editor

PLOS ONE

[Prof. Luis A. Gil-Alana gratefully acknowledges financial support from the PID2023-149516NB-I00/ AEI/10.13039/501100011033/ FEDER, UE project from ‘Ministerio de Economía, Industria y Competitividad’ (MINEIC), ‘Agencia Estatal de Investigación’ (AEI) Spain and ‘Fondo Europeo de Desarrollo Regional’ (FEDER), and also from Internal Projects of the Universidad Francisco de Vitoria.].

3. In the online submission form, you indicated that [The data are available from the authors upon request.].

Additional Editor Comments (if provided):

Reviewers' comments:

Reviewer's Responses to Questions

**Comments to the Author**

1. Is the manuscript technically sound, and do the data support the conclusions?

Reviewer #1: Partly

Reviewer #2: Yes

2. Has the statistical analysis been performed appropriately and rigorously?

Reviewer #1: No

Reviewer #2: Yes

3. Have the authors made all data underlying the findings in their manuscript fully available?

Reviewer #1: Yes

Reviewer #2: Yes

4. Is the manuscript presented in an intelligible fashion and written in standard English?

Reviewer #1: No

Reviewer #2: Yes

Reviewer #1: The manuscript presents an important and timely economic study. However, important technical and interpretation issues must be addressed before it can be recommended for publication. Revisions are needed to correct mathematical inaccuracies, strengthen methodological arguments, improve clarity, and align conclusions with the evidence.

\textbf{Recommendation:} Major Revision. Additional review comments are provided in the attached annotated PDF.

Reviewer #2: This paper offers a methodologically sound and policy-relevant contribution to price dynamics in the agricultural sector. With minor revisions to include more economic interpretation, visual data summaries, and connections to recent literature, it could serve as a valuable reference for both academic and applied economists.

**Do you want your identity to be public for this peer review?** For information about this choice, including consent withdrawal, please see our Privacy Policy

Reviewer #1: **Yes:** Kazeem Babatunde Akande

Reviewer #2: No

---

## [Decision Letter · Decision Letter 1]

20 Oct 2025

Dear Dr. Gil-Alana,

Thank you for submitting your manuscript to PLOS ONE. After careful consideration, we feel that it has merit but does not fully meet PLOS ONE’s publication criteria as it currently stands. Therefore, we invite you to submit a revised version of the manuscript that addresses the points raised during the review process.

We look forward to receiving your revised manuscript.

Kind regards,

Muhammad Umer Arshad

Academic Editor

PLOS ONE

Journal Requirements:

Additional Editor Comments:

The title refers to “structural breaks,” but no formal test for structural breaks has been performed; therefore, I suggest adjusting the title accordingly. Create a clearer connection between the methodological equations and the research objectives, and include a brief justification for choosing the FCVAR model over other common approaches. Remove repetition, shorten long sentences, and correct minor typographical errors (for example, Larg (Cost per Doze) should be Large (Cost per Dozen)). Also, check the notation in Equation (5) and ensure it uses a standard equation font.

Reviewers' comments:

Reviewer's Responses to Questions

**Comments to the Author**

Reviewer #3: All comments have been addressed

2. Is the manuscript technically sound, and do the data support the conclusions?

Reviewer #3: No

3. Has the statistical analysis been performed appropriately and rigorously?

Reviewer #3: No

4. Have the authors made all data underlying the findings in their manuscript fully available?

Reviewer #3: No

5. Is the manuscript presented in an intelligible fashion and written in standard English?

Reviewer #3: Yes

Reviewer #3: 1. Significance

The paper examines an important and policy-relevant issue — the persistence and mean-reverting behavior of average egg prices in the United States and their co-movement with the Producer Price Index (PPI). Food price stability is a crucial macroeconomic and welfare concern. The topic is both timely and valuable, especially given the high volatility of food prices during recent supply shocks. The application of fractional integration and FCVAR models is appropriate and potentially insightful.

2. Originality

The study’s originality lies in applying fractional integration and FCVAR approaches to egg price dynamics - an area that has received limited econometric attention. However, the contribution would be stronger if the authors explicitly position their study against existing literature on food price persistence that uses ARIMA/SARIMA or volatility models. The paper should better articulate how its methodological approach adds value beyond conventional time-series methods.

3. Methodology

Strengths

• The use of fractional integration and FCVAR is methodologically consistent with the study’s focus on persistence and long-memory behavior.

• Frequency-domain causality testing (Breitung and Candelon, 2006) is a sound preliminary step before fractional cointegration analysis.

Major Technical Issues

1. Estimation Details Missing:

The paper does not specify how the differencing parameter d was estimated (Whittle, MLE, semiparametric, etc.), nor the software used. Estimation procedures, confidence intervals, and replication details must be clearly stated.

2. Structural Breaks Not Tested:

Although the title mentions Structural Breaks, no formal test (Bai-Perron, Zivot-Andrews, or breaks in d) is presented. This omission undermines the title and conclusions. The authors should either include such tests or revise the title and discussion accordingly.

3. Seasonality Handling:

Monthly data are likely to contain seasonal components (e.g., holiday demand). While the paper mentions a seasonal AR(1) model, it does not clarify how seasonality was treated. The authors should discuss the choice of specification and its effect on the persistence parameter.

4. Residual Diagnostics:

The results depend on three different noise specifications (white noise, Bloomfield, seasonal AR(1)), but no diagnostic tests (Ljung–Box, ARCH tests, residual plots) are reported. Residual diagnostics are required to justify the final model choice.

5. Inconsistent Interpretation of d:

At some points, the paper claims “no evidence of short memory (I(0)),” yet elsewhere describes mean reversion (0.5 ≤ d < 1). The authors should unify their terminology and interpret d consistently throughout.

6. Data Availability and Transparency:

Although the data are said to be public, there is no DOI or repository link. Please provide direct data sources (e.g., FRED codes) and, if possible, an open-access CSV and code appendix for reproducibility.

7. Robustness Checks:

Consider re-estimating d using alternative semiparametric estimators (e.g., GPH, local Whittle) and provide bootstrap confidence intervals to ensure robustness.

4. Presentation and Structure

Language:

The manuscript is generally clear but would benefit from professional English editing to correct minor grammatical and typographical issues (e.g., inconsistent capitalization and minor spelling errors).

Tables:

Tables 2–5 should include the number of observations, AIC/BIC values, and explanatory notes for each column.

Figures:

Figures 2–3 need clearer axes labels, time scales, and detailed captions including data sources.

5. References

The references are relevant but could be expanded to include recent applications of fractional cointegration and structural break tests (e.g., Bai & Perron, 2003; Robinson, 1995; Velasco, 1999). Updating these will enhance the paper’s scholarly depth.

6. Major and Minor Revisions

Major Revisions Required

1. Include formal structural break tests (Bai-Perron or Zivot-Andrews) and discuss how breaks affect d and cointegration results.

2. Provide full estimation details — algorithms, software, sample size, convergence criteria, and replication code if possible.

3. Conduct robustness checks using alternative fractional estimators (e.g., GPH, local Whittle) and bootstrap confidence intervals.

4. Re-examine seasonality treatment and justify the chosen specification.

5. Add residual diagnostics and present key test statistics.

6. Provide data and code availability via a repository or appendix.

Minor Revisions

Improve figure and table clarity.

Conduct minor language editing.

Clarify the policy implications of findings (what does mean reversion imply for policymakers?).

**Do you want your identity to be public for this peer review?** For information about this choice, including consent withdrawal, please see our Privacy Policy

Reviewer #3: **Yes:** Hassan Tawakol Ahmed Fadol

---

## [Author Response · Author response to Decision Letter 2]

24 Oct 2025

Reviewers' comments:

Reviewer's Responses to Questions

Comments to the Author

1.) If the authors have adequately addressed your comments raised in a previous round of review and you feel that this manuscript is now acceptable for publication, you may indicate that here to bypass the “Comments to the Author” section, enter your conflict of interest statement in the “Confidential to Editor” section, and submit your "Accept" recommendation.

Reviewer #3: All comments have been addressed.

2.) Is the manuscript technically sound, and do the data support the conclusions?

Reviewer #3: No. See comments below.

3.) Has the statistical analysis been performed appropriately and rigorously?

Reviewer #3: No. See comments below.

4.) Have the authors made all data underlying the findings in their manuscript fully available?

Reviewer #3: No. See comments below.

5.) Is the manuscript presented in an intelligible fashion and written in standard English?

Reviewer #3: Yes .

6.) Review Comments to the Author

Reviewer #3:

1. Significance

The paper examines an important and policy-relevant issue — the persistence and mean-reverting behavior of average egg prices in the United States and their co-movement with the Producer Price Index (PPI). Food price stability is a crucial macroeconomic and welfare concern. The topic is both timely and valuable, especially given the high volatility of food prices during recent supply shocks. The application of fractional integration and FCVAR models is appropriate and potentially insightful.

Thank you for this positive comment.

2. Originality

The study’s originality lies in applying fractional integration and FCVAR approaches to egg price dynamics - an area that has received limited econometric attention. However, the contribution would be stronger if the authors explicitly position their study against existing literature on food price persistence that uses ARIMA/SARIMA or volatility models. The paper should better articulate how its methodological approach adds value beyond conventional time-series methods.

Thanks for this relevant comment. We have added the following paragraph in the literature review section:

“ … There are numerous studies that have focused the attention in the modelization of food prices throughout different techniques. Thus, for example, Mao et al. (2020) investigated China’s vegetable prices by using ARIMA models in order to determine the factors that influence vegetable price fluctuations. Jayatri (2025) used a more extended ARIMA-GARCH model to examine the volatility in various essential food prices in Indonesia. In a similar vein, Magnogna et al. (2025) employed a hybrid long short-term memory model for the analysis of agricultural price volatility in India. Other studies have performed empirical studies on price forecasting of individual agricultural prices such as soybean (Yin and Wang, 2021), pork (Wang and Sun, 2021; Ye et al., 2021), corn (Xu and Zhang, 2021) and beef (Zeng et al., 2019). Nevertheless, in the economic literature, the statistical properties of the average egg price have received limited attention, …”.

References:

Manogna, R.L., Dharmaji, V. & Sarang, S. (2025), A novel hybrid neural network-based volatility forecasting of agricultural commodity prices: empirical evidence from India. Journal of Big Data 12, 85. https://doi.org/10.1186/s40537-025-01131-8

Mao, L, Y. Huang, X. Zhang, S. LI and X. Huang (2020), ARIMA model forecasting analysis of the prices of multiple vegetables under the impact of the COVID-19, PLoS ONE 17(7): e0271594. https://doi.org/10.1371/journal.pone.0271594

Xu, X.J. and Y. Zhang (2021), Corn cash price forecasting with neural networks. Computers and Electronics in Agriculture,184, 106120

Ye K., Piao, Y.H.R., Zhao, K. and X.H. Cui (2021), A Heterogeneous Graph Enhanced LSTM Network for Hog Price Prediction Using Online Discussion. Agriculture,11(4) 359; https://doi.org/10.3390/agriculture11040359.

Yin, T and Y.M. Wang (2021), Market Efficiency and Nonlinear Analysis of Soybean Futures. SustainabilitY13(2), 518, https://doi.org/10.3390/su13020518.

Wang, C.S. and Z.H. Sun (2021),Monthly pork price forecasting method based on Census X12-GM(1,1) combination model.[J]. Plos one,16(5). https://doi.org/10.1371/journal.pone.0251436

Zeng, B., S.L. Li, W. Meng and D.H. Zhang (2019), An improved gray prediction model for China’s beef consumption forecasting. Plos one,14(9), pmid:31490952

3. Methodology

Strengths

• The use of fractional integration and FCVAR is methodologically consistent with the study’s focus on persistence and long-memory behavior.

• Frequency-domain causality testing (Breitung and Candelon, 2006) is a sound preliminary step before fractional cointegration analysis.

Thank you!

Major Technical Issues

1. Estimation Details Missing:

The paper does not specify how the differencing parameter d was estimated (Whittle, MLE, semiparametric, etc.), nor the software used. Estimation procedures, confidence intervals, and replication details must be clearly stated.

Thanks for the comment. We have added the following in the methodological section:

“ … The estimation relies on the parametric approach of Robinson (1994), which is a testing procedure based on the Lagrange Multiplier (LM) principle and that uses the likelihood function in the frequency domain. We use a very simple approach of this method that has been widely used in the empirical literature (see Gil-Alana and Robinson, 1997 for the specific functional form of the tests used in this application, whose codes can be found in Gil-Alana, 1998). The main features of this procedure are …”.

Also,

“ …“ … Data and codes used in this application are available in the following repository 10.5281/zenodo.17415399 at https://zenodo.org/uploads/17415399. …”.

References

Gil-Alana, L.A. (1998), Testing fractional integration in macroeconomic time series. PhD thesis, London School of Economics and Political Science, London, UK.

Gil-Alana, L.A. and P.M. Robinson, (1997). Testing of unit root and other nonstationary hypotheses in macroeconomic time series. Journal of Econometrics, 80(2), 241-268. doi:10.1016/s0304-4076(97)00038-9

Robinson, P. M. (1994). Efficient Tests of Nonstationary Hypotheses. Journal of the American Statistical Association, 89(428), 1420–1437. https://doi.org/10.2307/2291004

2. Structural Breaks Not Tested:

Although the title mentions Structural Breaks, no formal test (Bai-Perron, Zivot-Andrews, or breaks in d) is presented. This omission undermines the title and conclusions. The authors should either include such tests or revise the title and discussion accordingly.

Thanks for letting us know this point. We have proceeded accordingly, removing the presence of “breaks” in the title, and extending a comment in the conclusions about potential lines of future research. The new title is now as follows:

“Average Egg Price: Mean reversion and Persistence in a Time Series Approach”.

We have also added the following comment in the conclusions:

“ … The paper contains some limitations. First, the possibility of structural breaks is an issue that has not been investigated in the present work. This is relevant since long memory and fractional integration are closely related to the issue of breaks (see, Granger and Hyung, 2008; Barassi et al., 2018; etc.). In this context, the methodology proposed in Gil-Alana (2008), which is basically an extension of Bai and Perron’s (2003) approach to the fractional case can be implemented to the same data used in this application. …”.

References

Bai, J. and P. Perron (2003), Computation and analysis of multiple structural change models, Journal of Applied Econometrics 18, 1, 1-22.

Barassi, M.R., Spagnolo, N. & Zhao, Y. (2018), Fractional Integration Versus Structural Change: Testing the Convergence of Emissions. Environ Resource Econ 71, 923–968. https://doi.org/10.1007/s10640-017-0190-z

Gil-Alana, L.A. (2008), Fractional integration and structural breaks at unknown periods of time, Journal of Time Series Analysis 29, 1, 163-185.

Granger, C.W.J. and N. Hyung (2004), Occasional structural breaks and long memory with an application to the S&P 500 absolute stock returns, Journal of Empirical Finance 11, 3, 399-421.

3. Seasonality Handing:

Monthly data are likely to contain seasonal components (e.g., holiday demand). While the paper mentions a seasonal AR(1) model, it does not clarify how seasonality was treated. The authors should discuss the choice of specification and its effect on the persistence parameter.

This is another relevant point that was not clearly explained in the old manuscript. We have added the following:

“ … Note that seasonality does not appear as a relevant issue in any of the two series investigated since the estimates of d are practically identical under the seasonal AR specification and the white noise form for the error term. In fact, though unreported, the seasonal AR coefficients were statistically insignificant in the two cases. …”.

4. Residual Diagnostics:

The results depend on three different noise specifications (white noise, Bloomfield, seasonal AR(1)), but no diagnostic tests (Ljung–Box, ARCH tests, residual plots) are reported. Residual diagnostics are required to justify the final model choice.

Thanks for the comment. Note that this cannot be performed in the context of the model of Bloomfield (1973) since this is a non-parametric approach with no explicit functional form. However, performing tests of no autocorrelation in the other two specifications, the results supported the view of no additional time dependence.

“ … It should finally be noted that several Ljung-Box Q-test statistics were conducted on the residuals from the white noise and seasonal AR(1) specifications and the results suggested that they were no additional serial correlation in the data. …”.

References

Bloomfield, P. (1973). An exponential model in the spectrum of a scalar time series, Biometrika 60, 217-226.

5. Inconsistent Interpretation of d:

At some points, the paper claims “no evidence of short memory (I(0)),” yet elsewhere describes mean reversion (0.5 ≤ d < 1). The authors should unify their terminology and interpret d consistently throughout.

Thanks for the comment. We refer to short memory or I(0) behaviour if the hypothesis of d = 0 is rejected in the data. This happens if that value does not belong to the 95% confidence intervals reported in the tables. Mean reversion takes place if d is smaller than 1. We have clarified these points all over the manuscript.

In this context, we have changed two sentences in the Abstract, that reads now as follows:

“This study investigates the long-term behavior of average egg prices in the U.S., using fractional integration techniques. The analysis also explores the relationship between egg prices and the Producer Price Index (PPI) through cointegration methods. The findings suggest long memory in both series and mean reversion for PPI. Additionally, the FCVAR model indicates a potential long-run equilibrium relationship between average egg prices and the PPI.”.

Also, in the conclusions we have made this point much more explicit:

“ … The primary discovery of this paper supports the notion that disturbances in average egg prices will have enduring effects in the U.S., as none of the series displays short memory traits (i.e., I(0)). Thus, long memory takes place in all cases with positive values of d across all confidence intervals. This suggests that abrupt changes or fluctuations in prices are likely to have a more prolonged impact. Moreover, the fact that the intervals include the value of d = 1 indicates that we cannot reject the hypothesis of a unit root in the series and therefore we may reject the hypothesis of reversion to the mean. On the other hand, regarding the Producer Price Index, despite the absence of short memory traits, the parameter d falls within the range 0.5 ≤ d < 1. This indicates that the time series is nonstationary with mean reversion,

6. Data Availability and Transparency:

Although the data are said to be public, there is no DOI or repository link. Please provide direct data sources (e.g., FRED codes) and, if possible, an open-access CSV and code appendix for reproducibility.

We have added the following comment:

“ … We use a very simple approach of this method that has been widely used in the empirical literature (see Gil-Alana and Robinson, 1997 for the specific functional form of the tests used in this application, whose codes can be found in Gil-Alana, 1998). …”.

Also, in the data section:

“ … Data and codes used in this application are available in the following repository 10.5281/zenodo.17415399 at https://zenodo.org/uploads/17415399. …”.

References

Gil-Alana, L.A. (1998), Testing fractional integration in macroeconomic time series. PhD thesis, London School of Economics and Political Science, London, UK.

Robinson, P. M. (1994). Efficient Tests of Nonstationary Hypotheses. Journal of the American Statistical Association, 89(428), 1420–1437. https://doi.org/10.2307/2291004

7. Robustness Checks:

Consider re-estimating d using alternative semiparametric estimators (e.g., GPH, local Whittle) and provide bootstrap confidence intervals to ensure robustness.

Thanks for this comment. We have included the following comment at the end of the Methodological section:

“ … Nevertheless, as a robustness check, we also employed alternative approaches like the parametric method of Sowell (1992) and the semiparametric log-periodogram estimate of Geweke and Porter-Hudak (1983) and others like Robinson (1995) and Velasco (1999), all producing very similar results to those reported in this work. …”.

References

Geweke, J. & Porter-Hudak, S. (1983), The estimation and application of long memory time series. Journal of Time Series Analysis 4(4), 221-238. https://doi.org/10.1111/j.1467-9892.1983.tb00371.x

Robinson, P. M. (1995), Gaussian semiparametric estimation of long range dependence, Annals of Statistics 23, 1630-1661.

Sowell, F. (1992), Maximum likelihood estimation of stationary univariate fractionally integrated time series models, Journal of Econometrics 53(1-3), 165-188. https://doi.org/10.1016/0304-4076(92)90084-5

Velasco, C. (1999), Gaussian Semiparametric Estimation of Non-stationary Time Series, Journal of Time Series Analysis 20, 1, 87-127.

8. Presentation and Structure

Language:

The manuscript is generally clear but would benefit from professional English editing to correct minor grammatical and t

---

## [Editor Report · Decision Letter 2]

29 Oct 2025

Dear Dr. Gil-Alana,

Thank you for submitting your manuscript to PLOS ONE. After careful consideration, we feel that it has merit but does not fully meet PLOS ONE’s publication criteria as it currently stands. Therefore, we invite you to submit a revised version of the manuscript that addresses the points raised during the review process.

We look forward to receiving your revised manuscript.

Kind regards,

Muhammad Umer Arshad

Academic Editor

PLOS ONE

Journal Requirements:

Additional Editor Comments:

The abstract is too short. it doesn’t explain the importance of the study, the data used, or what the results mean. Please revise and expand to clearly say what question the paper answers, what data period was used, what methods were applied, and what the main findings mean in simple terms. Table 1 and 5 the headings for tables are too brief. they should be more descriptiv, please revise. In addition, the conclusion is too long and repeats earlier parts of the paper. it should be shorter and only focus on what was found, why it matters, and what could be done in future work.

---

## [Author Response · Author response to Decision Letter 3]

30 Oct 2025

Additional Editor Comments:

The abstract is too short. it doesn’t explain the importance of the study, the data used, or what the results mean. Please revise and expand to clearly say what question the paper answers, what data period was used, what methods were applied, and what the main findings mean in simple terms. Table 1 and 5 the headings for tables are too brief. they should be more descriptiv, please revise. In addition, the conclusion is too long and repeats earlier parts of the paper. it should be shorter and only focus on what was found, why it matters, and what could be done in future work.

Reply: Dear Editor, Many thanks for these additional comments. Following your suggestions, we have first modified the Abstract that now reads as follows:

“ This study investigates the long-term behavior of average egg prices and the Producer Price Index (PPI) in the U.S., using fractional integration and cointegration techniques. Using monthly data from 1980 to 2023, the results indicate evidence of long memory in both series and mean reversion for PPI. Thus, exogenous shocks in the PPI series will return by themselves to the original values. Additionally, there is a long-run equilibrium relationship between average egg prices and the PPI.”.

The heading in Tables 1 and 5 have been extended. Thus,

Table 1: Descriptive statistics of Average Egg Prices and PPI Egg Prices

Table 5: Results of cointegration between Average Egg Prices and PPI using the Fractional Cointegration VAR (FCVAR) approach

Finally, the conclusions section has been reduced, removing several paragraphs that were repetitions and no much informative for the purpose of the work. Thus, the new section is only formed by three paragraphs: one explaining the issue investigated, a second one displaying the main results, and a final one with the limitation of the present work:

“The objective of this paper has been to analyze the stochastic behavior of the average egg price (cost per dozen) in the U.S. by looking at the statistical properties of the series and using fractional integration. Additionally, the study aimed to understand the possible long run relationship between PPI and the average egg price.

The results indicate that for the average egg price, we observe lack of reversion to the mean, implying permanency of shocks and requiring policy actions (e.g., subsidy design or inventory stabilization) in case of shocks to recover the original trends. Conversely, for the Producer Price index data, mean reversion seems to take place in all cases, making it useless for policymakers to take action when sudden peaks occur. Furthermore, both the correlation coefficient and the causality test prove the existence of some a positive relationship between the two variables. Furthermore, the results obtained using the FCVAR model indicate that the average egg price and the producer price index are cointegrated with the cointegrating errors displaying an I(0) behaviour. These findings suggest the presence of a stable long-run equilibrium relationship between the Average Egg price and the Producer Price Index in the U.S., indicating a sustained co-movement between the two variables over time.

The paper contains some limitations. First, the possibility of structural breaks is an issue that has not been investigated in the present work. This is relevant since long memory and fractional integration are closely related to the issue of breaks (see, Granger and Hyung, 2008; Barassi et al., 2018; etc.). In this context, the methodology proposed in Gil-Alana (2008), which is basically an extension of Bai and Perron’s (2003) approach to the fractional case can be implemented to the same data used in this application. In addition, non-linear trends can replace the linear structure employed in this work. Other alternative long memory model can also be used.”.

Hoping this time the paper will be finally accepted in PlosOne,

Yours sincerely,

The authors

---

## [Editor Report · Decision Letter 3]

2 Nov 2025

Dear Dr. Gil-Alana,

Thank you for submitting your manuscript to PLOS ONE. After careful consideration, we feel that it has merit but does not fully meet PLOS ONE’s publication criteria as it currently stands. Therefore, we invite you to submit a revised version of the manuscript that addresses the points raised during the review process.

We look forward to receiving your revised manuscript.

Kind regards,

Muhammad Umer Arshad

Academic Editor

PLOS ONE

Journal Requirements:

Additional Editor Comments:

The author has attempted to revise the abstract, but it still needs further improvement. I recommend reviewing abstracts from similar published papers to better understand how to clearly present the study’s purpose, key findings, and significance.

---

## [Author Response · Author response to Decision Letter 4]

3 Nov 2025

Additional Editor Comments:

The author has attempted to revise the abstract, but it still needs further improvement. I recommend reviewing abstracts from similar published papers to better understand how to clearly present the study’s purpose, key findings, and significance.

Reply: Thanks for the comment. See the new Abstract:

The evolution over time of the Consumer Price Index (CPI) is regarded as a key indicator of the general health and direction of any given economy. As the CPI continues to rise, the purchasing power of consumers decreases and their spending habits change significantly, making it imperative for policymakers to understand the underlying reasons that lead to such changes. A key component of the CPI basket is represented by the food and beverages items, within which eggs have undergone a significant price increase during the past years. Egg prices have a significant impact on consumers, given that eggs are a staple product, serving as the lowest cost protein alternative. This paper analyzes the long term behavior of the average egg price (cost per Dozen) in the U.S. by looking at the statistical properties of the series and using a methodology based on the concept of fractional integration. The primary goal is to determine whether the average egg price exhibits traits of long memory or mean reversion. Long memory describes the scenario where observations from a distant past have an influence on the present value of the series. Conversely, mean reversion refers to the phenomenon where data points eventually return to the long-term average after deviating from the mean for a certain period. The analysis also explores the relationship between egg prices and the Producer Price Index (PPI) through cointegration methods. Preliminary findings indicate that long memory takes place in both series and mean reversion in the PPI. Also, the two series seem to be cointegrated. This suggests the presence of a stable long-run equilibrium relationship between the Average Egg price and the Producer Price Index in the U.S., indicating a sustained co-movement between the two variables over time.

---

## [Editor Report · Decision Letter 4]

5 Nov 2025

Average Egg Price: Mean reversion and Persistence in a Time Series Approach

PONE-D-25-28658R4

Dear Dr. Gil-Alana,

We’re pleased to inform you that your manuscript has been judged scientifically suitable for publication and will be formally accepted for publication once it meets all outstanding technical requirements.

Kind regards,

Muhammad Umer Arshad

Academic Editor

PLOS ONE
---

## [Editor Report · Acceptance letter]

PONE-D-25-28658R4

PLOS One

Dear Dr. Gil-Alana,

I'm pleased to inform you that your manuscript has been deemed suitable for publication in PLOS One. Congratulations! Your manuscript is now being handed over to our production team.

Kind regards,

on behalf of

Dr. Muhammad Umer Arshad

Academic Editor

PLOS One